# G-SPEED: General SParse Efficient Editing MoDel

**Haoke Zhang,**[*] **Yue Wang,**[*] **Juntao Li,**[†] **Xiabing Zhou, Min Zhang**
Soochow University
{hkzhangnlp, ywangnlp}@stu.suda.edu.cn
{ljt, zhouxiabing, minzhang}@suda.edu.cn

## Abstract

Large Language Models (LLMs) have demonstrated incredible capabilities in understanding, generating, and manipulating languages. Through human-model interactions, LLMs can automatically understand human-issued instructions and output the expected contents, which can significantly increase working efficiency. In various types of real-world demands, editing-oriented tasks account for a considerable proportion, which involves an interactive process that entails the continuous refinement of existing texts to meet specific criteria. Due to the need for multi-round human-model interaction and the generation of complicated editing tasks, there is an emergent need for efficient general editing models. In this paper, we propose **G**eneral **SP**arse **E**fficient **E**diting Mo**D**el (**G-SPEED**), which can fulfill diverse editing requirements through a single model while maintaining low computational costs. Specifically, we first propose a novel unsupervised text editing data clustering algorithm to deal with the data scarcity problem. Subsequently, we introduce a sparse editing model architecture to mitigate the inherently limited learning capabilities of small language models. The experimental outcomes indicate that G-SPEED, with its 508M parameters, can surpass LLMs equipped with 175B parameters. Our code and model checkpoints are available at https://github.com/Banner-Z/G-SPEED.

## 1 Introduction

Recently, the Natural Language Processing (NLP) community has witnessed the rapid development of Large Language Models (LLMs). With the help of instruction tuning, LLMs can effectively utilize the knowledge acquired during pre-training and have the ability to follow user instructions (Wei et al., 2021; Sanh et al., 2021; Mishra et al., 2022; Wang et al., 2022; Ouyang et al., 2022; OpenAI, 2023).

---

[*] Equal Contribution
[†] Corresponding Author

---

Therefore, due to the powerful ability of large models as assistants, users can conveniently accomplish a wide range of open-domain tasks by collaborating with LLMs. One particularly important applied scenario of the LLM assistants is editing.

Editing is a crucial process in writing, requiring a diverse range of skills to refine texts from multiple perspectives. Consequently, there is a long-term goal of using machines to assist in automating the editing process. However, the inherent difficulty of editing, coupled with the limited availability of annotated editing data, hinders the development of general editing models capable of meeting a wide range of editing requirements. Therefore, before the era of LLMs, previous works mainly focused on developing specialized editing models tailored to specific tasks, such as grammatical error correction (GEC) (Bryant et al., 2019), style transfer (Rao and Tetreault, 2018), and sentence fusion (Geva et al., 2019). Due to their specific capabilities, these specialized editing models have limited practical applications. Besides, although recent work has explored the potential of LLMs as general editing models (Faltings et al., 2021; Schick et al., 2023; Madaan et al., 2023), the iterative nature of editing often requires multiple interactions with models to edit a single piece of text. Therefore, relying solely on LLMs to meet diverse editing needs becomes impractical and cost-prohibitive.

In this work, to develop a lightweight general edit model that caters to various editing requirements, we propose **G**eneral **SP**arse **E**fficient **E**diting Mo**D**el (**G-SPEED**). Specifically, we introduce a novel unsupervised editing data clustering method to eliminate noise and annotate the intent, considering the limited quality of current edit intent annotations and the presence of noisy data in the revision history of Wikipedia passages. Besides, we propose a novel sparse editing model structure to enhance the learning capabilities of small models. Experimental results demonstrate that G-SPEED

achieves state-of-the-art performance on the EditEval benchmark and exhibits a strong generalization ability to unseen tasks with limited data resources.

In conclusion, our contributions are as follows:

- In this work, we first propose G-SPEED, a lightweight framework with a novel sparse editing model structure, which can satisfy various editing needs;

- To deal with the limited editing data source, we propose a novel unsupervised data clustering method, which will be released to promote the development of editing models;

- Experimental results show that G-SPPED can achieve state-of-the-art results on EditEval, surpassing the best LLMs by 2.8 points on average, and have a strong generalization ability to unseen tasks in data-limited scenarios.

## 2    Related work

**General Text Editing**    Existing work on general text editing can broadly be divided into two categories: (1) instruction-based text editing (Faltings et al., 2021; Schick et al., 2023; Madaan et al., 2023);(2) multi-intent editing (Du et al., 2022b; Kim et al., 2022). Instruction-based text editing polishes up existing texts following user instructions, which is based on LLMs (Chung et al., 2022; Brown et al., 2020; Ouyang et al., 2022) and needs a long time to generate and large-scale datasets for training. Multi-intent editing uses intents to perform diverse editing actions. Specifically, Kim et al. (2022) improves text editing with intent span detection. However, edit intents are defined by humans, which have high annotation costs and may not cover all editing intents (Yang et al., 2017; Anthonio et al., 2020; Du et al., 2022b). In this work, we propose G-SPEED, which can handle general text editing without the dependence on LLMs and human-annotate editing intents.

**Editing Model**    Editing models are efficient alternatives to Seq2Seq models (Sutskever et al., 2014) when the source and target texts in the task have a large amount of overlap, e.g., grammatical error correction (GEC) (Bryant et al., 2019), style transfer (Rao and Tetreault, 2018), and sentence fusion (Geva et al., 2019). Editing models learn to predict edit operations and directly leave the correct text unchanged, while Seq2Seq models generate target text from scratch. LaserTagger (Malmi et al.,

2019) is a general approach that predicts edit operations by sequence labeling and then generates inserted words with a fixed vocabulary. Omelianchuk et al. (2020) proposes more complete tags (such as pluralization and capitalization) to make generation easier and improve the GEC task. Mallinson et al. (2020) and Mallinson et al. (2022) perform arbitrary content insertion via non-autoregressive and semi-autoregressive methods, respectively. However, the current editing model is stuck on solving a single task. Editing models do not perform well in multi-task problems (Du et al., 2022b).

**Sparse Language Model**    Sparse language models achieve promising results with larger model sizes while maintaining almost the same computational costs (Jaszczur et al., 2021; Du et al., 2022a; Fedus et al., 2022b,a). There are various routing algorithms to determine where to send examples, such as hash routing (Roller et al., 2021), base layer (Lewis et al., 2021), and Multi-gate MoE (Ma et al., 2018). Zuo et al. (2022), Gupta et al. (2022) and Lee-Thorp and Ainslie (2022) design sparse feed-forward layers on BERT (Devlin et al., 2019) and make a boost on GLUE (Wang et al., 2018). In this paper, we explore the use of sparse layers on Editing models for General Text Editing tasks with a BERT backbone.

## 3    Task Formulation

During text editing, the documents $D$ undergo modifications based on user-selected intents $I$ like fluency, clarity, simplification, and neutralization. Each modification corresponds to a pair of documents $(D_{t-1}, D_t)$ and an editing intent $I_t$ where $t$ represents the number of modifications. Text editing models are tailored to modify documents based on intent:

$$D_t = f(D_{t-1}, I_t), \qquad (1)$$

where $f$ denotes text editing models.

## 4    Unsupervised Editing Data Clustering

To facilitate text editing tasks, a dataset comprising editing intent and text pairs, both before and after editing, is required for pre-training purposes. ITERATER (Du et al., 2022b) collects text revisions from Wikipedia, ArXiv, and Wikinews. It categorizes text intents into five categories: fluency, clarity, coherence, style, and meaning-changed. The intent is then predicted using a RoBERTa-based model (Liu et al., 2019) that takes both the original

| Cluster | Source | Target | User Comment |
|---|---|---|---|
| Fluency | At the end of the 1986 season, he announced that would retire after completing the 1987 NFL season. | At the end of the 1986 season, he announced that he would retire after completing the 1987 NFL season. | Minor grammatical fix |
| Readability | The journey to London takes approx. 53 minutes. | The journey to London takes about one hour. | 'about one hour' is more readable than 'approx. 57 min.' |
| Simplification | European sales were slowed when 150,000 faulty CD copies of the album were recalled by record company EMI. Discs sent to Germany, France, Spain, Portugal, the Netherlands and Belgium had been affected by a mastering error. CDs started with a 40 second live recording of a different band - Pearl Jam - according to a fan site. | 150,000 European CD copies of the album were recalled by EMI after a mastering error was discovered. | rm unnecessary detail |
| Neutralization | CORBA aims to bring to the table many benefits that no other single technology brings in one package. | CORBA supports several features which it claims that no other single technology brings in one package. | more neutrality. Let's unwrap this rhetorical language |

Table 1: Representative examples in our pre-training dataset.

and revised texts as input. However, automatic annotation performs poorly on small categories due to the unbalanced distribution of categories, and certain categories, such as simplification, are not taken into account. Formulating perfect intent categories and annotating large-scale data for pre-training pose significant challenges.

Similar to existing work (Zhang et al., 2019; Faltings et al., 2021), we first collect data from the revision histories in March 1st, 2023 dump of English Wikipedia.[1] Each revision consists of an original text, a revised text, and a user comment that is associated with the editing intent. We extract revisions from the XML format dump and exclude revisions that do not have user comments. To ensure data clustering quality, we additionally perform data cleaning based on various factors, such as the BLUE value of the source and target, as well as the length of comments and text. We extract the source and target sentences using *Punkt Sentence Tokenizer*[2] and *difflib*[3], and then we eliminate Wikipedia markup using *wikiextractor*[4].

To classify user comments into distinct intent categories, we utilize $k$-means clustering initialized with $k$-means++ algorithm (Arthur and Vassilvitskii, 2007).[5] With a collection of $n$ comment

embeddings represented as $x_1, x_2, ...x_n$, $k$-means clustering aims to partition these embeddings into $k$ sets $\{S_1, S_2, ...S_k\}$ so as to minimize the intra-cluster distance:

$$\min_S \sum_{i=1}^{k} \sum_{x \in S_i} dist(x, \mu_i), \quad (2)$$

where $dist$ represents the Euclidean distance, and $\mu_i$ denotes the mean of embeddings within $Si$.

In our study, we utilize Sentence-BERT (Reimers and Gurevych, 2019) to generate comment embeddings, which are then decomposed using Singular Value Decomposition (SVD). Sentence-BERT provides a more effective representation of text semantics compared to algorithms based on word frequency.

To discern the purpose of each cluster, we also organize the designed prompts into clusters (such as fixing grammar errors and improving text cohesion). Based on the prompts they contained, we then selected four clusters, namely fluency, readability, simplification, and neutralization, as illustrated in step I of Figure 1. Table 1 presents the data instances. Clusters that focus on information updates, citation modifications, punctuation modifications, and other similar tasks are disregarded. For more details, please refer to Appendix A.

## 5 G-SPEED

### 5.1 Editing Model

We decompose editing into two steps: (1) **tagging**, label each word with an editing operation by sequence labeling; (2) **generation**, insert new words into the original sentence. As shown in Figure 1, we share the encoder parameters of tagging and

---

[1] https://dumps.wikimedia.org/enwiki/
[2] https://www.nltk.org/api/nltk.tokenize.punkt.html#module-nltk.tokenize.punkt
[3] https://docs.python.org/3/library/difflib.html#module-difflib
[4] https://github.com/attardi/wikiextractor
[5] We also employ the DBSCAN algorithm (Ester et al., 1996), which segments categories by detecting changes in data point density and eliminates the need to specify the number of clusters. However, due to the high and uneven density of data points, DBSCAN struggles to segment meaningful clusters effectively.

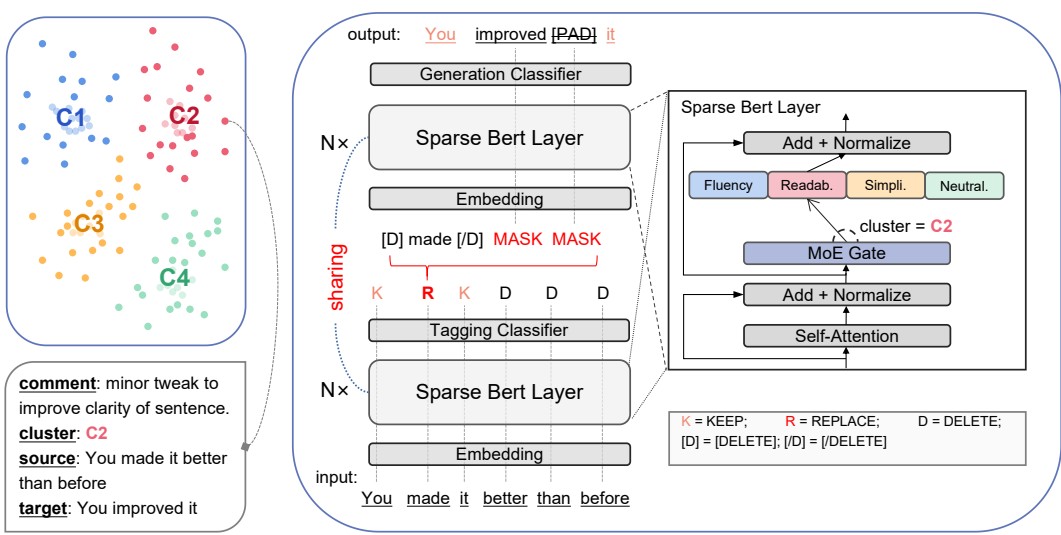

Figure 1: The illustration of unsupervised editing data clustering and G-SPEED training. Initially, we utilize unsupervised data clustering by leveraging user comments. Subsequently, sparse BERT layers are utilized for training G-SPEED with multi-task data, wherein the feed-forward layer associated with the respective cluster is activated. Text editing is divided into two distinct steps: initially predicting the operation tags and subsequently inserting the words. The encoders for the two steps are shared. "Readab." means readability, "Simpli." denotes simplification, and "Neutral." means neutralization.

generation so that the two modules are trained together:

$$\mathcal{L} = \mathcal{L}_{tagging} + \lambda\mathcal{L}_{generation} \qquad (3)$$

where $\lambda$ is a hyper-parameter.

**Tagging** Editing operations are marked at the word level. A linear classification layer is used to predict tags following a transformer encoder. Training is performed using a cross-entropy loss:

$$\mathcal{L}_{tagging} = -\sum_{i}^{|x^t|} \log p(y_i^t | f_{cls}(h_i^t)), \qquad (4)$$

where $y^t$ is golden tags, $i$ is the index of each word, $h^t$ is the output of encoder, $x^t$ is the input text, $|x^t|$ represents the total number of words in the text, $f_{cls}$ is a classification layer to predict tags.

The most basic type of editing operation is represented by the Levenshtein transition types, which encompass *KEEP*, *DELETE*, *REPLACE*, and *APPEND*. Omelianchuk et al. (2020) expands the sum of types to 4971, 29 of which represent fundamental text transformations, and the rest are used to insert new words. To maximize the number of tags for reducing generation difficulties while ensuring sufficient training data for each tag, we develop

a total of 14 tags which encompass case conversion, singular and plural conversion, verb tense conversion, as well as split operations like hyphen splitting or merging. For automatic operation annotation, we utilize dynamic programming and set the editing distance as the cost function. Detailed information about the categories and a comparison of various designs are provided in Appendix B.

**Generation** The tagging phase primarily handles transformations that do not involve the generation of new words. During the generation phase, our task is to predict new words at the positions of *REPLACE*, *APPEND*, and *TRANSFORM-VERB*. We employ an efficient non-autoregressive mask language model (MLM) similar to Mallinson et al. (2020). As shown in Figure 1, we insert $n$ *[MASK]* tokens at each required position and subsequently perform mask prediction on the new text.[6] Any *[MASK]* tokens beyond the length of the desired tokens will be predicted as *[PAD]*. The MLM is trained with a cross-entropy loss:

$$\mathcal{L}_{generation} = -\sum_{i}^{|y^s|} \log p(y_i^s | f_{pred}(h_i^s)), \qquad (5)$$

where $y^s$ is the golden results of *[MASK]*, $i$ is the index of *[MASK]*, $|y^s|$ denotes the total number of

---

[6] In our experiments, $n = 4$.

*[MASK]*, $h^s$ is the hidden states, and $f_{pred}$ is linear layers to predict tokens in vocabulary.

To retain the maximum amount of information from the original sentence, we retain the deleted words and the verbs with the wrong tense. As demonstrated in Figure 1, we enclose the deleted words within *[DELETE]* and *[/DELETE]* tags, and the wrong verbs within *[TRANSFORM_VERB]* and *[/TRANSFORM_VERB]* tags.

## 5.2 Sparse Bert Layer

Then, we introduce an efficient and compact sparse Bert layer for a multi-intent editing model. Figure 1 illustrates the use of four fully connected feed-forward networks (FFN), referred to as "experts". The experts correspond to four editing intents present in our training data. The experts are activated when training data that corresponds to their respective intents. The data point shown in the figure is clustered as "C2" (readability) and, therefore, utilizes the red module. Each layer comprises 8 experts, with the tagging and generation modules being divided due to their distinct effects. By employing this approach, we can efficiently address multi-intent editing tasks with a single encoder:

$$h^t = f_\theta(x^t, r^t, z^t), r \in [0, n), z \in [0, 1], \quad (6)$$

where $x^t$ represents the input for either tagging or generation, $r^t$ represents the index of the intent, $n$ is the total number of intents, $z^t = 0$ indicates the tagging mode and $z^t = 1$ indicates the generation mode, $f_\theta$ represents the sparse Bert model, and $h^t$ is the hidden states in Equation 4 and 5.

For the balance between each task and the two modules, we sequentially train the tagging module and the generation module of each task in units of a batch, so that the number of training steps for each task and each module is the same.

## 5.3 Additional Fine-tune

In practice, text editing tasks may not align with our expert configuration. We employ additional fine-tuning for specific tasks. To enhance the model's generalization, we copy the experts in the pre-trained model and train targeted tasks accordingly. Specifically, we freeze all parameters except the experts, enabling the model to adapt flexibly to any editing task with minimal training cost.

## 6 Experiments

### 6.1 Datasets

We conduct our evaluation using EditEval (Dwivedi-Yu et al., 2022), which is an instruction-based text improvement benchmark. The study includes six datasets and six tasks; their details are provided below.

**Fluency** The primary objective of the fluency task is to rectify grammatical and spelling errors. The evaluation of this task utilizes ITERATER (Du et al., 2022b) and JFLEG (Napoles et al., 2017). ITERATER is an edit-intention annotated corpus of iteratively revised text. We utilize the subset that focuses on fluency. In comparison to ITERATER, JFLEG not only corrects grammatical errors but also makes the original text more native sounding.

**Clarity and Coherence** Both tasks use the corresponding subsets in the test dataset of ITERATER. The clarity task aims to enhance the formality, conciseness, readability, and comprehensibility of the text. The coherence task aims to enhance the cohesiveness, logical connectivity, and overall coherence of the text.

**Paraphrasing** The purpose of the Paraphrasing task is to rephrase sentences without a specific improvement goal. EditEval opts to utilize text pairs from SemEval-2018 (Cer et al., 2017) that have high similarity scores.

**Simplification** TurkCorpus (Xu et al., 2016) and ASSET (Alva-Manchego et al., 2020) are uesed for evaluation of this task. ASSET and TURK contain identical test examples, but their references are simplified in different ways.

**Neutralization** The process of neutralization involves eliminating subjective bias in a text. For example, in the text of "Jimi hendrix (musician), a great musician and vocalist", the word "great" lacks neutrality. The WNC dataset (Pryzant et al., 2020) serves as a representative dataset for this task.

### 6.2 Metrics

**SARI** (Xu et al., 2016) measures the goodness of words that are added, deleted, and kept. It calculates the average n-gram F1 score for addition, deletion, and retention operations. **GLUE** (Napoles et al., 2015) is an additional n-gram-based metric better suited for evaluating single sentences than BLEU. It is applied to JFLEG and ITERATER.

| Cluster (size) | Fine-tune | Size | Test | Abbrev. |
|---|---|---|---|---|
| Fluency (140K) | W&I+LOCNESS | 5K | JFLEG | JFL |
| | ITERATER-V2 | 5K | ITERATER | ITR-F |
| Readability (180K) | ITERATER-V2 | 10K | ITERATER | ITR-L |
| | ITERATER-V2 | 10K | ITERATER | ITR-O |
| | ParaSCI | 10K | SemEval-2018 | STS |
| Simplification (91K) | Turk | 2K | Turk | TRK |
| | Wikiauto | 8K | Asset | AST |
| Neutralization (93K) | WNC | 10K | WNC | WNC |

Table 2: Statistics of the pre-training dataset, fine-tuning dataset, and test set. "Abbrev." denotes the abbreviation of the test dataset. Readability corresponds to the three tasks of clarity, coherence, and paraphrasing.

**Exact Match** (EM) calculates the percentage of predictions that exactly match the reference. This metric is the official measure used by WNC.

### 6.3 Baselines

Initially, we assess our models using a non-pre-trained baseline, which employs identical fine-tuning data and edit model structure as G-SPEED. Then, we assess various existing LLMs using the EditEval setup, employing prompts to enable diverse editing operations. Our primary comparison is **PEER** (Schick et al., 2023), a collaborative language model trained on Wikipedia edit history that is initialized from the LM Adapted variant of T5 (Schick et al., 2023). Scores for 3b and 11b parameters PEER are reported on EditEval. **T0, T0++** (Sanh et al.) and T**k-Instruct** (Wang et al., 2022) are other models that are initialized from the LM Adapted variant of T5 and then fine-tuned using prompts or instructions. We compare against **GPT-3** (Brown et al., 2020), **OPT** (Zhang et al., 2022), **InstructGPT** (Ouyang et al., 2022), and **ChatGPT**[7] as decoder-only LLMs.

### 6.4 Implementation Details

We present the information of the datasets in Table 2. For additional fine-tuning, we select data from various sources: W&I+LOCNESS (Bryant et al., 2019), ParaSCI (Dong et al., 2021), Turk (Xu et al., 2016), Wikiauto (Jiang et al., 2020), WNC (Pryzant et al., 2020), and the subsets of ITERATER-V2 (Kim et al., 2022) with intent confidence scores higher than 0.8. Each expert has 10k data points for fine-tuning.

For data clustering, we employ K-Means++ with cuML (Raschka et al., 2020) toolkit, setting the number of clusters to 10. We choose all-mpnet-

base-v2[8] to generate comment embeddings. We use scikit-learn (Pedregosa et al., 2011) to employ SVD and set the output dimension to 100.

For pre-training, we choose bert-base-cased (Devlin et al., 2019)[9] as our backbone model with the Huggingface Transformers (Wolf et al., 2020) toolkit. We initialize all experts based on the corresponding counterparts in Bert. We use Adam Optimizer (Kingma and Ba, 2014) and clip the norms of gradients to 1. $\lambda$ is set to 1.

During additional fine-tuning, the expert responsible for the readability task is duplicated into three instances. Each instance is utilized to train the tasks of clarity, coherence, and paraphrasing, as shown in Table 2. All experiments are carried out on 4 Nvidia GeForce RTX 3090 GPUs.

### 6.5 Main Results

The main results are presented in Table 3. The metrics are compared among the following: Copy the text of source (**Copy**), supervised state-of-the-art models (**SotA**), **LLMs**, and our model (**G-SPEED**). The average SARI score is calculated as the mean of each task, while the score for a specific task is computed as the mean of the corresponding test datasets. Out of the models with fewer than 11 billion parameters, only PEER achieves a score more than 5 points higher than the average Copy score. Despite their large size, models such as OPT and GPT-3, which have 175B parameters, are unable to ensure satisfactory performance in general text editing tasks. Among the LLMs, InstructGPT performs the best, slightly greater than ChatGPT. Additionally, PEER demonstrates favorable results with a relatively small number of parameters.

G-SPEED surpasses the performance of LLMs with only 508M parameters. Compared with the best LLMs, InstructGPT and ChatGPT, G-SPEED achieves satisfactory results on coherence and neutralization tasks. G-SPEED exhibits a noticeable gap compared to ChatGPT in fluency, paraphrasing, and simplification tasks. However, these areas can be significantly improved through additional fine-tuning. We compare our model with two variants: one without the pre-training step (denoted as "w/o PRE.") and another without the fine-tuning step (denoted as "w/o ADDI.") as shown in Table 3. We find that both of these steps are crucial for our model. The model without either

---

[7]https://chat.openai.com/

[8]https://huggingface.co/sentence-transformers/all-mpnet-base-v2

[9]https://huggingface.co/bert-base-cased

| Method | Size | Fluency | | Clarity | Coherence | Para. | Simplification | | Neutral. | Avg. |
|---|---|---|---|---|---|---|---|---|---|---|
| | | JFL | ITR-F | ITR-L | ITR-O | STS | TRK | AST | WNC | |
| Copy | - | 26.7 / 40.5 | 32.3 / 86.0 | 29.5 / 62.9 | 31.3 / 77.2 | 21.1 | 26.3 | 20.7 | 31.9 / 0.0 | 27.8 |
| SotA | - | - / 62.4 | 37.2 / – | 46.2 / - | 38.3 / - | - | 34.4 | 37.2 | -/45.8 | - |
| T*k*-instruct | 3B | 31.8 / 39.0 | 32.4 / 61.6 | **38.4 / 58.4** | 33.8 / **70.4** | 30.2 | 32.8 | 29.9 | 31.3 / 0.4 | 32.9 |
| T0 | 3B | 42.0 / 38.8 | 24.6 / 34.9 | 32.6 / 30.2 | 22.2 / 21.6 | 34.3 | 34.4 | 32.3 | 22.3 / 0.0 | 29.7 |
| T0++ | 11B | 34.7 / 43.2 | 35.3 / 75.8 | 37.6 / 56.5 | 32.7 / 59.9 | 28.4 | 32.9 | 28.2 | 29.3 / 0.3 | 32.3 |
| PEER-3B | 3B | 55.5 / 54.3 | 51.4 / 84.3 | 32.1 / 47.1 | 32.1 / 59.8 | 28.6 | 32.5 | 30.5 | 53.3 / 21.6 | 38.5 |
| PEER-11B | 11B | 55.8 / 54.3 | **52.1 / 85.2** | 32.5 / 51.3 | 32.7 / 62.7 | 28.2 | 32.1 | 29.5 | 54.5 / 22.8 | 38.8 |
| OPT | 175B | 47.3 / 47.5 | 34.7 / 70.6 | 31.5 / 31.5 | 27.6 / 36.1 | 29.1 | 32.6 | 31.8 | 31.2 / 0.4 | 32.1 |
| GPT-3 | 175B | 50.3 / 51.8 | 32.1 / 56.7 | 33.5 / 39.7 | 26.9 / 36.1 | 27.2 | 33.0 | 30.5 | 31.7 / 0.6 | 32.0 |
| InstructGPT | 175B | 61.8 / 59.3 | 48.8 / 82.7 | 35.1 / 48.4 | 35.9 / 60.2 | **42.5** | 38.8 | 38.0 | 35.4 / 2.2 | 40.4 |
| ChatGPT | - | **65.6 / 61.3** | 45.6 / 70.4 | 31.5 / 30.5 | 33.7 / 33.5 | 36.9 | 39.7 | **44.6** | 34.1 / 0.0 | 39.0 |
| G-SPEED (Ours) | 508M | 54.2 / 51.9 | 42.2 / 78.8 | 34.0 / 53.6 | **41.0** / 68.4 | 36.0 | 38.5 | 40.6 | **60.2 / 31.2** | **43.2** |
| w/o PRE. | 508M | 48.4 / 47.1 | 36.6 / 77.2 | 34.1 / 53.3 | 40.2 / 70.1 | 33.6 | **39.8** | 40.3 | 57.7 / 28.8 | 41.4 |
| w/o ADDI. | 508M | 45.6 / 46.6 | 43.6 / 79.3 | 36.9 / 53.9 | 36.8 / 65.5 | 29.4 | 34.4 | 37.1 | 60.1 / 30.9 | 40.6 |

Table 3: The performance of G-SPPED and all baseline models on EditEval. The best score among LLMs and G-SPEED is **Bold**, and the second is Underlined. "Para." denotes paraphrasing, "Neutral." means neutralization, and "Avg." stands for the average SARI score for each task. The first numbers for each task are SARI scores; additional metrics are GLEU for fluency, clarity, and coherence, and EM for neutralization. "w/o PRE." means the model fine-tuned without pre-training, and "w/o ADDI." denotes the pre-trained model without additional fine-tuning.

step performs poorly. In particular, the pre-training step greatly improves fluency, clarity, and neutralization tasks, while the fine-tuning step enhances coherence, paraphrasing, and simplification tasks. This can be attributed to the varying proportions of these tasks in the history of Wikipedia editing. Furthermore, our model, even without additional fine-tuning, performs slightly better than InstructGPT and ChatGPT, thereby confirming the effectiveness of our training and clustering method.

## 6.6 Further Analysis

**Expert Structure**   In Table 4, we present the results for two types of expert structures: (1) sparse last encoder layer (referred to as "Last Layer"), which often impacts the final output, and (2) sparse feed-forward layer (referred to as "Feed Forward") in each encoder layer. Additionally, we include the results of a dense model (referred to as "Dense"), where the entire encoder is shared, and differentiation between different tasks occurs solely at the linear classification layer. Our findings indicate that sparse models generally outperform the dense model, with inference times being almost identical. Furthermore, models with sparse feed-forward layers show slightly better performance than those with sparse last encoder layers. Furthermore, we examine a model that shares the experts in tagging and generation (referred to as "Sharing T&G"), which can further reduce the model's size. However, this approach yields inferior performance compared to the other models.

**Router Structure**   We compare one static routing algorithm and two dynamic routing algorithms. (1) Select the expert corresponding to the task (referred to as "Task ID"). (2) Select the expert using a linear classification layer (referred to as "Linear"), which predicts the probabilities by softmax for each expert and multiplies the highest probability with the output of the corresponding expert. (3) Select the expert using a linear classification layer specific to each task (referred to as "Task ID + Linear"), which differs only in the number of classifiers with "Linear".[10]   As shown in Table 4, the routing algorithms combined with task information perform better. We believe this is because the editing intention cannot be directly inferred from the input text. The external knowledge about editing intentions benefits model training. Furthermore, we employ a "Token Level" routing algorithm in our model, which selects the experts for each token based on the hidden states of the token. However, this approach yields inferior performance. Selecting an expert based on the semantics of a whole sentence is more reasonable and efficient.

**Few Shot Learning**   Table 5 presents the results of few-shot learning using the G-SPEED backbone for general text editing tasks. We maintain the same settings as in the additional fine-tuning, setting the data volume for each task at 500, 1k, 5k, and 10k, respectively. We compare the

---

[10]The router weights are initialized using a normal distribution with a mean of 0 and a standard deviation of 0.001. The temperature of the softmax in the router is 0.7.

| Method | Fluency | | Clarity | Coherence | Para. | Simplification | | Neutral. | Avg. |
|---|---|---|---|---|---|---|---|---|---|
| | **JFL** | **ITR-F** | **ITR-L** | **ITR-O** | **STS** | **TRK** | **AST** | **WNC** | |
| **Dense** | 42.8 / 46.0 | 42.0 / 79.6 | 35.5 / **55.3** | 36.8 / 67.8 | 24.6 | 32.6 | 35.7 | 57.5 / 27.8 | 38.5 |
| **Last Layer** | | | | | | | | | |
| Task ID | 44.8 / 47.2 | 42.6 / 80.4 | 36.1 / 53.4 | 36.9 / 65.6 | 28.3 | 32.7 | 37.1 | 59.1 / 28.5 | 39.8 |
| Linear | 43.1 / 46.2 | 42.5 / 79.9 | 36.0 / 54.0 | 37.7 / 67.9 | 26.3 | 34.0 | 37.4 | 58.1 / 28.0 | 39.4 |
| Task ID + Linear | 45.0 / **47.3** | 41.8 / 79.7 | 35.9 / 54.2 | **39.1** / 67.9 | 28.6 | 33.0 | 37.1 | 59.0 / 28.4 | 40.2 |
| Sharing T&G | 40.0 / 44.3 | 40.8 / **80.7** | 34.7 / 54.8 | 36.9 / **69.2** | 24.5 | 33.4 | 37.4 | 56.9 / 26.3 | 38.1 |
| **Feed Forward** | | | | | | | | | |
| Task ID | **45.6** / 46.6 | **43.6** / 79.3 | 36.9 / 53.9 | 36.8 / 65.5 | 29.4 | 34.4 | 37.1 | **60.1 / 30.9** | **40.6** |
| Sharing T&G | 43.1 / 46.0 | 42.7 / 80.4 | 35.5 / 54.5 | 36.0 / 68.1 | 26.7 | 33.7 | 37.4 | 57.1 / 27.0 | 39.0 |
| Linear | 44.6 / 47.0 | 42.9 / 79.9 | 36.5 / 54.6 | 37.8 / 68.4 | 28.5 | 32.4 | 36.5 | 58.8 / 28.2 | 40.0 |
| Token Level | 44.2 / 46.7 | 41.4 / 79.1 | 36.1 / 54.7 | 35.6 / 64.0 | **29.6** | 31.9 | 37.6 | 59.3 / 29.2 | 39.8 |
| Task ID + Linear | 45.3 / 46.8 | 41.2 / 78.3 | **37.0** / 54.7 | 38.0 / 68.1 | 28.6 | **35.0** | **37.7** | 59.3 / 29.8 | 40.4 |
| Token Level | 45.0 / 44.4 | 40.2 / 77.0 | 36.8 / 52.4 | 36.5 / 65.0 | 28.2 | 34.3 | 37.6 | 59.0 / 30.4 | 39.8 |

Table 4: Comparison of various sparse layers on EditEval. The best score is **Bold**, and the second is Underlined. "Linear" means a linear classifier for routing; "Task ID + Linear" denotes a classifier specific to each task; "Sharing T&G" means using the same experts for tagging and generation model; "Token Level" indicates that experts are activated separately for each token.

| Method | Train Data Size | | | |
|---|---|---|---|---|
| | **0.5K** | **1K** | **5K** | **10K** |
| G-SPEED | **41.4** | **41.8** | **42.4** | **42.9** |
| w/o PRE. | 38.9 | 39.5 | 40.8 | 41.4 |

Table 5: Comparison of our pre-trained and non-pre-trained models in few-shot learning. The head of the column is the amount of training data for each task. We report the average SARI score for each task.

| Method | 0.1% | 0.01% |
|---|---|---|
| BERT2BERT(Rothe et al., 2020) | 3.4 | 0.0 |
| T5-base(Raffel et al., 2020) | 33.8 | 10.8 |
| LASERTAGGER (Malmi et al., 2019) | 25.7 | 12.3 |
| FELIX (Mallinson et al., 2020) | 36.9 | 17.0 |
| EDIT5 (Mallinson et al., 2022) | 43.8 | 28.6 |
| G-SPEED (Ours) | **47.8** | **33.2** |

Table 6: Comparison of Seq2Seq models, single task editing models, and G-SPEED on sentence fusion task (Exact Match) under various data conditions.

model that skips the pre-training step (referred to as "w/o PRE") with our pre-trained model. We find that the pre-trained model outperforms the non-pre-trained model. Furthermore, the fine-tuning results with 500 samples are nearly equivalent to those of the un-pre-trained model trained on 10k samples, demonstrating its generalization ability on downstream tasks. Additionally, we fine-tune our pre-trained model on the sentence fusion task and compare it with editing models. Following the work of Mallinson et al. (2022), we use the "balanced Wikipedia" subset of the DiscoFuse dataset (Geva et al., 2019) and compare the results with those of editing models using 4,500 (0.1%) and 450 (0.01%) training data points. We report the Exact Match score of G-SPEED, while the other results are reported by Mallinson et al. (2022, 2020). In Table 6, our initial observation is that the Seq2Seq model (BERT2BERT (Rothe et al., 2020), T5 base (Raffel et al., 2020)) performs poorly, indicating that editing models are effective on train-

ing data. Furthermore, fine-tuning our pre-trained model yields significantly better performance than other editing models, demonstrating the generalization ability of our method on downstream tasks.

## 7 Conclusion

In this work, we propose the General SParse Efficient Editing MoDel (G-SPEED), a model designed to fulfill diverse editing needs while ensuring computational efficiency. Specifically, we first propose a novel unsupervised clustering strategy to obtain a large amount of multi-intent editing data, which is collected from editing histories from Wikipedia and can be used to conduct pre-training. Subsequently, we propose a novel sparse editing model architecture to improve the learning abilities of small models. The experimental results on EditEval show that, with the use of Bert-base as the backbone model, G-SPEED can outperform LLMs while maintaining an efficient inference speed. Ad-

ditionally, we discuss different sparse structures and show the strong generalization capability of our method across downstream tasks.

## Limitations

A limitation of unsupervised clustering is that it cannot deal with revisions that contain more than one editing intent. Although text editing in the history of Wikipedia is iterative, the degree of each modification by the user is still uncontrollable.

## Ethics Statement

We collect all data from publicly available sources and test our model on public datasets. Since our work mainly focuses on non-meaning-changing text edits, we are able to avoid many issues involving generating harmful text. Therefore, our work has no possibility of generating harmful information in practical applications.

## Acknowledgements

This work is supported by the National Science Foundation of China (No. 62176174, No. 62206194) and the Natural Science Foundation of Jiangsu Province (No. BK20220488).

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

## A  Details and Studies on Datasets

In addition to the filtering rules discussed in Section 4, we employ additional filtering criteria for revisions. Firstly, we exclude revisions whose comments contain any of the following terms: "template", "image", "infobox", "pic", "link", "photo", "comment", "http:", "https:", ".jpg", ".png", or "reply". The presence of these terms suggests the involvement of unwanted revision types, such as image alterations and citation modifications, which can complicate the clustering process (e.g., due to lengthy URL links in comments). Furthermore, we remove any links present in the source and target texts. Additionally, we replace certain important shortcuts with their full meanings to enhance data clustering. For instance, we substitute "[[WP:NPOV|POV]]" with "neutral point of view", "[[WP:TYPO]]" with "typo", "[[WP:RS]]" with "reliable sources", and "[[WP:SYN]]" with "synthesis". This modification contributes to improved clustering outcomes. Furthermore, we exclude revisions that solely pertain to number or time modifications, such as updates on the real-time number of COVID-19 cases in a specific location. Moreover, we filter out revisions in which both the comments and the source texts exhibit significant similarities. This type of overlap revision typically corresponds to low-quality comments. Since our extraction of source and target relies solely on document comparison, there are instances where we may encounter incorrect or incomplete source and target pairs. Consequently, it becomes necessary to conduct a basic screening of the clustered data. One approach involves utilizing BLUE to identify and filter out sentence pairs that do not match, thereby minimizing the presence of inaccuracies. Additionally, another filtering criterion involves considering differences in sentence length to identify and exclude incorrect simplification data.

Table 2 presents the central words and a selection of sampled comments within each cluster. A concise description is provided for each cluster. The clusters employed for training purposes are clusters 1, 2, 3, and 5. The central words correspond to the most frequently occurring words in each cluster, excluding stop words, and they exhibit a high degree of concentration within their respective clusters. Detailed information regarding the dataset instances is presented in Table 1. Figure 2 illustrates the distribution of semantic clustering result maps in a two-dimensional space. To enhance the visualization of clustering effects, we employed a random subsample from the dataset and applied the Uniform Manifold Approximation and Projection (UMAP) technique to condense the semantic data into a two-dimensional vector representation.

We have undertaken a meticulous examination to determine the presence of any data contamination between our training and test datasets. Specifically, in the evaluation benchmark, JFLEG, Turk, ASSET, and ITERATER are manually annotated; STSB is collected from forums and news. Only the WNC dataset is collected from the revision history of Wikipedia passages. However, WNC is collected from the revisions made between 2004 and 2019, while our dataset is collected from the revision histories in the March 1st, 2023 dump of English Wikipedia. For further confirmation, we first concatenate the source and target of each data sample and use traversal search to filter the training samples of our pre-training datasets that have a BLUE value of 0.9 with any data sample of WNC. However, no such data sample satisfies this requirement, which further confirms there is no contamination.

## B  Details and Experiments on Tags

Similar to the approach taken in GEC-TOR (Omelianchuk et al., 2020), our work considers fine-grained tag design. We present three different tag designs in Table 8, where the tag names and meanings correspond to those used in GECTOR. The decision to reduce the number of tags from 34 to 14 is based on the frequency of tag occurrence in the pre-training dataset and its compatibility with the pre-training process. Table 9 displays the performance results of the three

| Cluster | Center Words | Comment Samples | Description |
|---|---|---|---|
| 0 | born, father, wife, died, birth, death | "+A little history on the Spanish Monastery"; "Source for Jane Merriett Netterville party and year of birth"; | Supplementation and modification of information about character relationships, character information, history, etc. |
| 1 | structure, rewrote, copy-edit, tweaks, flow, rewording, readability, awkward, improve, clarity, phrasing | "reworded for clarity"; "paragraph split, wording"; "Fixing awkward sentence"; "Small sentence change to make sentence seem more natural."; "fix awkward phrasing."; "Made more clear and precise."; | Modification of sentence structure and wording, improvement of readability and clarity, and copy-edit |
| 2 | neutral, view, opinion, evidence, true, point, claim | "and skepticism, the other main scientific bias"; "it was controversial"; "1. More accurate info. 2. No proof or source that anyone lost their savings."; "removed bias and point of view pushing"; | Neutralization, verification of evidence, and removal of bias and confusion |
| 3 | cleanup, typos, fixing, errors, density, mistakes, misspelling, grammatical | "Typo fixing, replaced: anatomy anatomy Ž192 anatomy"; "Clean up, typo fixed: the USA Ž192 the US"; "minor grammatical change" | Correction of grammar, typo, misspelling and other errors |
| 4 | updating, updated, add, info | "Add date with source."; "Addition of storyline info."; "adding page and info about copyright"; | Information updates and additions |
| 5 | deletion, vandalism, duplicate, delete, redundant, superfluous | "Removing Hurrian deities, who have a separate category down below. Inara is not Hurrian, neither is Hannahanna. Also, Hebat was listed multiple times."; | Text simplification, information removal |
| 6 | units, numbers, %, = | "In units where c=1, mass and energy are expressed in the same units - e.g. in Kg."; "Mexican Navy orders 2 units."; "It's only the second aircraft operated." | Messy statements |
| 7 | citations, cite, references, ref, reliable | "new inline reference"; "help please - similar problem, but the cite button has disappeared"; "Added needed citations." | Correction and verification of references |
| 8 | capitalized, draft, plural, hyphen, capitalization | "Hyphenation and consistency"; "what is with the capital letters?"; | Modification of hyphenation, capitalization, etc. |
| 9 | italicize, comma, italics, semicolon, dashes, colon, marks, parenthesis | "putting in ( ) marks to make the page better"; "changed bold font into quotes"; "changed hyphen to colon"; | Modification of punctuation and font |

Table 7: The center words and comments of each cluster with a description.

| Sum | Tags |
|---|---|
| 4 (KDRA) | KEEP, DELETE, REPLACE, APPEND |
| 14 (ours) | APPEND, DELETE, KEEP, MERGE_HYPHEN, REPLACE, TRANSFORM_AGREEMENT_PLURAL, TRANSFORM_AGREEMENT_SINGULAR, TRANSFORM_CASE_CAPITAL, TRANSFORM_CASE_CAPITAL_1, TRANSFORM_CASE_LOWER, TRANSFORM_CASE_UPPER, TRANSFORM_CASE_UPPER_-1, TRANSFORM_SPLIT_HYPHEN, TRANSFORM_VERB |
| 34 | APPEND, DELETE, KEEP, MERGE_HYPHEN, MERGE_SPACE, REPLACE, TRANSFORM_AGREEMENT_PLURAL, TRANSFORM_AGREEMENT_SINGULAR, TRANSFORM_CASE_CAPITAL, TRANSFORM_CASE_CAPITAL_1, TRANSFORM_CASE_LOWER, TRANSFORM_CASE_UPPER, TRANSFORM_CASE_UPPER_-1, TRANSFORM_SPLIT_HYPHEN, TRANSFORM_VERB_VBD_VB, TRANSFORM_VERB_VBD_VBG, TRANSFORM_VERB_VBD_VB, TRANSFORM_VERB_VBD_VBZ, TRANSFORM_VERB_VBG_VB, TRANSFORM_VERB_VBG_VBD, TRANSFORM_VERB_VBG_VBN, TRANSFORM_VERB_VBG_VBZ, TRANSFORM_VERB_VBN_VB, TRANSFORM_VERB_VBNVBD, TRANSFORM_VERB_VBN_VBG, TRANSFORM_VERB_VBN_VBZ, TRANSFORM_VERB_VBZ_VB, TRANSFORM_VERB_VBZ_VBD, TRANSFORM_VERB_VBZ_VBG, TRANSFORM_VERB_VBZ_VBN, TRANSFORM_VERBVB_VBD, TRANSFORM_VERB_VB_VBG, TRANSFORM_VERB_VB_VBN, TRANSFORM_VERB_VB_VBZ |

Table 8: Details of the tag sets we compared.

| Tag Design | Fluency | | Clarity | Coherence | Para. | Simplification | | Neutral. | Avg. |
|---|---|---|---|---|---|---|---|---|---|
| | JFL | ITR-F | ITR-L | ITR-O | STS | TRK | AST | WNC | |
| **Last Layer** | | | | | | | | | |
| 14tags (Ours) | 45.0 / 47.3 | 41.8 / 79.7 | 35.9 / 54.2 | **39.1 / 67.9** | 28.6 | 33.0 | 37.1 | 59.0 / 28.4 | 40.2 |
| KDRA | 44.6 / 46.8 | 42.9 / **79.9** | 35.8 / 54.1 | 37.5 / 67.3 | 29.2 | 32.8 | 36.9 | 58.9 / 28.6 | 40.0 |
| 34tags | 45.3 / 46.9 | 42.6 / 78.8 | 36.6 / 53.4 | 37.1 / 66.2 | 29.7 | 33.4 | 37.1 | 58.7 / 28.2 | 40.2 |
| **Feed Forward** | | | | | | | | | |
| 14tags (Ours) | 45.6 / 46.6 | **43.6** / 79.3 | 36.9 / 53.9 | 36.8 / 65.5 | 29.4 | 34.4 | 37.1 | **60.1** / 30.9 | **40.6** |
| KDRA | **45.9** / 47.3 | 41.9 / 78.5 | **37.0** / 54.2 | 36.1 / 65.4 | 28.6 | 34.4 | 37.1 | 58.9 / 29.6 | 40.0 |
| 34tags | 45.3 / 46.6 | 41.1 / 78.2 | 36.4 / 54.0 | 35.5 / 64.2 | **30.3** | **34.5** | 37.1 | 59.4 / 30.9 | 40.1 |

Table 9: Comparison of the impact of different tag designs on the models.

Figure 2: UMAP projection of the training dataset. The four colors represent the four clusters used in pre-training.

| LaserTagger | FELIX | G-SPEED |
|---|---|---|
| 33.0 | 33.5 | 43.2 |

Table 10: Comparison of three editing models that were trained with the same dataset. The results in the table are average SARI scores across multiple tasks.

tag designs on two sparse models. Notably, the design comprising 14 tags achieves the highest performance among all designs.

## C   Efficiency Comparison

As demonstrated in Table 10, G-SPEED exhibits superior multi-tasking capabilities when contrasted with previous editing models due to its sparse architecture. Furthermore, as shown in Table 11, compared to fine-tuned Large Language Models (LLMs), G-SPEED boasts conspicuous advantages in model size, computational speed, and performance. Specifically, we use the WNC test set for inference on a single NVIDIA GeForce RTX 3090. We conducted fine-tuning on both T0 and LLaMa using the identical dataset. We also adjusted the data input format to "instruction: input," as was performed during instruction fine-tuning. Notably,

| Model | Model Size ↓ | SARI ↑ | Tokens/s ↑ | Speed Up ↑ |
|---|---|---|---|---|
| LLaMa-7B | 1.00x | 54.1 | 21.05 | 1.00x |
| T0-3B | 0.43x | 30.5 (-23.6) | 39.46 | 1.87x |
| G-SPEED (508M) | **0.07x** | **60.2 (+6.1)** | **754.33** | **35.84x** |

Table 11: Comparison between fine-tuned LLMs and G-SPEED.

| Model | SARI | GLEU |
|---|---|---|
| ChatGPT | 36.4 | 30.9 |
| G-SPEED | 36.6 | **48.3** |

Table 12: Comparison of multi-intent iterative text modification capabilities.

G-SPEED achieves a remarkable speed boost up to 35 times faster than LLaMa-7B. Moreover, the training costs of G-SPEED are substantially more economical than those of LLMs.

# D   Iteration Study

There is currently no large test set for multi-intent iterative editing tasks. In our study, we employed the iterator_human_doc dataset[11], which comprises a total of 51 manually annotated instances, to assess and contrast the iterative editing proficiency of both ChatGPT and G-SPEED. Table 12 shows that the two models are similar in the SARI score, but G-SPEED retains the original text better.

We also explored multiple iterations within a single task. As indicated in Table 13, we conducted a comparison of our pre-trained model at various editing depths. Our findings reveal that the fluency task consistently benefits from iterative editing, whereas most tasks are more inclined toward single-pass modifications. Table 14 presents specific instances from the JFLEG dataset, illustrating the incremental revision process employed by the editing model to address grammatical errors.

---

[11] https://huggingface.co/datasets/wanyu/
IteraTeR_human_doc/viewer/default/test

| Method | Fluency | | Clarity | Coherence | Para. | Simplification | | Neutral. |
|---|---|---|---|---|---|---|---|---|
| | JFL | ITR-F | ITR-L | ITR-O | STS | TRK | AST | WNC |
| **Last Layer** | | | | | | | | |
| depth = 1 | 45.0 / 47.3 | 41.8 / **79.7** | 35.9 / **54.2** | **39.1 / 67.9** | 28.6 | **33.0** | **37.1** | **59.0 / 28.4** |
| depth = 2 | 48.7 / 49.4 | 42.5 / 79.1 | 36.1 / 53.9 | 37.7 / 65.7 | 28.6 | 28.4 | 36.0 | 55.5 / 12.6 |
| depth = 3 | **49.5 / 49.7** | **42.6** / 78.7 | 36.1 / 54.1 | 37.5 / 65.6 | **29.1** | 24.6 | 33.2 | 54.7 / 12.8 |
| **Feed Forward** | | | | | | | | |
| depth = 1 | 45.6 / 46.6 | 43.6 / **79.3** | **36.9 / 53.9** | **36.8 / 65.5** | 29.4 | **34.4** | **37.1** | **60.1 / 30.9** |
| depth = 2 | 49.6 / 48.7 | **43.7** / 78.3 | 36.3 / 53.2 | 35.7 / 63.2 | 29.6 | 30.0 | 35.8 | 54.8 / 8.8 |
| depth = 3 | **50.6** / 48.9 | 41.4 / 76.4 | 36.6 / 53.0 | 35.0 / 60.5 | **30.2** | 26.3 | 33.4 | 53.3 / 9.4 |

Table 13: Comparison of the results after multiple iterations. **Bold** indicates the best score in each model.

| Depth | Outputs |
|---|---|
| 0 | Bigger farming are use more chemical product and substance to feed fish . |
| 1 | Bigger farming uses more chemical product and substance to feed fish. |
| 2 | Bigger farming uses more chemical products and substances to feed fish. |
| 3 | Bigger farming uses more chemical products and substances to feed fish. |
| 0 | they did not get the ideas or any concepts about what they learn . |
| 1 | They did not get the ideas or any concepts about what they learn. |
| 2 | They did not get the ideas or any concepts about what they learned. |
| 3 | They did not get the ideas or any concepts about what they learned. |
| 0 | I larned many kind of subject , also I could make difrent types friends . |
| 1 | I studied many kind of subject, also I could make different types friends. |
| 2 | I studied many kind of subjects, also I could make different types of friends. |
| 3 | I studied many kind of subjects, also I could make different types of friends. |

Table 14: The iteration cases on JFLEG. The depth of 0 refers to the source text.