# OpenReview forum: "G-SPEED: General SParse Efficient Editing MoDel"
_EMNLP/2023/Conference — EMNLP 2023 Findings_

### Official Review · Reviewer_VHDk · 2023-08-04

**Typos Grammar Style And Presentation Improvements:** 1. In Section "Related work (General …
**Soundness:** 2

**Excitement:**

3: Ambivalent: It has merits (e.g., it reports state-of-the-art results, the idea is nice), but there are key weaknesses (e.g., it describes incremental work), and it can significantly benefit from another round of revision. However, I won't object to accepting it if my co-reviewers champion it.

**Paper Topic And Main Contributions:**

The article proposes a general text editing model called G-Speed. The authors pretrain the model using text editing data collected from Wikipedia, arXiv, and Wikinews, to alleviate data sparsity issues. To enable the model to handle multiple editing tasks simultaneously, the authors employ a mixture-of-experts architecture for training. Results on six datasets demonstrate that G-SPEED achieves superior performance.

**Questions For The Authors:**

1. Why does Table 6 evaluate the model's performance only on the sentence fusion task?
2. In Table 3, the performance of w/o ADDI. outperforms G-SPEED on ITR-F and ITR-L. Please explain the reason for this discrepancy.

Question for Rebuttal

I've carefully reviewed the authors' thoughtful responses and the additional experimental results have certainly addressed part of my concerns. However, I still have some concerns regarding the contributions of this paper:

(1) The authors claim that prior work is specifically designed for a particular type of editing task. However, PEER is proposed to address a variety of editing tasks and has leveraged data from Wikipedia to effectively mitigate the issue of data scarcity in text editing. Furthermore, the results in Table 3 demonstrate that PEER still performs competitively in a zero-shot setting across multiple editing tasks.

(2) Additionally, as mentioned in Lines 061-067, some existing work based on LLMs has already achieved general text editing capabilities. The authors emphasize the importance of efficiency in text editing, particularly in interactive scenarios. However, the main experiments in the paper focus only on single-turn editing and do not provide any inference time comparisons between different models.  Moreover, these LLM-based methods can edit texts based on user feedback to cover a broader range of editing needs, making them more practical. Consequently, I think that the experimental results do not adequately support the authors' motivation.

**Reasons To Accept:**

1. The paper presents a model capable of handling various text editing tasks with superior performance on multiple tasks.
2. The unsupervised edit data clustering algorithm can effectively leverage text editing data from real-world platforms.

**Reasons To Reject:**

1. Insufficient experimental comparisons.
a) Most of the main baselines are evaluated in zero-shot setting, while G-SPEED is further fine-tuned with supervised task data, leading to an unfair comparison. It is suggested to include more supervised baseline models, such as PEER with further fine-tuning.
b) The evaluation metrics may not be fully appropriate. Consider adding human evaluation or automated evaluation with ChatGPT/GPT4.
c) In Table 6, the comparison involves G-SPEED and other SeqSeq models trained with only a small amount of supervised data. However, the selected models in Table 6 are not pre-trained with massive edit data, and it is essential to compare G-SPEED with the supervised models (T0, PEER).
2. The contribution needs further elaboration. While the paper claims that they propose an unsupervised edit data clustering method to mitigate data scarcity problem (Lines 021-023), previous work (PEER) already utilize edit data from Wikipedia to train a general text editing model. The advantages of the proposed algorithm over prior work are not clearly articulated.
3. Text editing often involves multi-turn interactions, but the proposed model is only tested for single-turn editing requests. Evaluating text editing performance under multi-turn settings, possibly using human evaluation, would make the experimental results more convincing.

**Reproducibility:**

3: Could reproduce the results with some difficulty. The settings of parameters are underspecified or subjectively determined; the training/evaluation data are not widely available.

**Reviewer Confidence:**

3: Pretty sure, but there's a chance I missed something. Although I have a good feel for this area in general, I did not carefully check the paper's details, e.g., the math, experimental design, or novelty.

---

> ### Author Rebuttal · Authors · 2023-08-29
>
> Thank you so much for the informative reviews and inspiring comments on our paper! We hope the following responses can address your questions well.
>
> > Reject Reason 1: Insufficient experimental comparisons. a) Most of the main baselines are evaluated in zero-shot setting, while G-SPEED is further fine-tuned with supervised task data, leading to an unfair comparison. It is suggested to include more supervised baseline models, such as PEER with further fine-tuning. b) The evaluation metrics may not be fully appropriate. Consider adding human evaluation or automated evaluation with ChatGPT/GPT4.
> A: Thanks for your suggestions.
>
> a) In Table 3, we have reported the performance of G-SPEED without fine-tuning("w/o ADDI"). We can observe that G-SPEED can still outperform all the baselines without further fine-tuning. Since the datasets and models of PEER are not open source so far, we choose to use downstream supervised data to fine-tune LLaMa-7B with LoRA to further confirm the effectiveness of G-SPEED. The results are as follows:
>
> |          | JFL      | ITR-F    | ITR-L    | ITR-O    | STS      | TRK  | AST      | WNC      |
> | -------- | -------- | -------- | -------- | -------- | -------- | ---- | -------- | -------- |
> | LLaMa-7B | 52.0     | 36.3     | 33.9     | 36.8     | **36.3** | 38.5 | 37.1     | 54.1     |
> | G-SPEED  | **54.2** | **42.2** | **34.0** | **41.0** | 36.0     | 38.5 | **40.6** | **60.2** |
>
> The results show that G-SPEED performs better than LLaMa-7B when both conduct additional fine-tuning.
>
> b) We conduct human evaluation between G-SPEED and ChatGPT. The following are the results of human evaluation:
>
> | ChatGPT win | Tie   | G-SPEED win |
> | ----------- | ----- | ----------- |
> | 28.3%       | 40.3% | 31.4%       |
>
> The results of the human evaluation show that G-SPEED outperforms ChatGPT. The next version will report these results and more human evaluation details.
>
> We will report these results in the next version.
>
> > Reject Reason 2: The contribution needs further elaboration.
>
> A: Due to the data scarcity problem of editing tasks, there is a line of existing works exploring utilizing the revision history of Wikipedia passages. All these methods are either introduced for specific editing tasks or designed for LLMs. In our preliminary study, both of these methods fail to achieve the goal of achieving an efficient general editing model. Therefore, we think the efficient general editing model has a much higher requirement for training data quality than the specific editing model and general editing LLMs. Due to the challenge of obtaining an efficient general editing model, we propose a novel unsupervised editing data clustering method, which can achieve higher data quality than existing methods.
>
>
>
> > Reject Reason 3: Text editing often involves multi-turn interactions, but the proposed model is only tested for single-turn editing requests. Evaluating text editing performance under multi-turn settings, possibly using human evaluation, would make the experimental results more convincing.
>
> A: We tested the interactive modification in IteraTeR_human_doc and compared ChatGPT. The results are as follows:
>
> |         | SARI     | GLEU     |
> | ------- | -------- | -------- |
> | ChatGPT | 36.4     | 30.9     |
> | G-SPEED | **36.6** | **48.3** |
>
> The results show that the two models have similar modification capabilities, but G-SPEED retains the source text more.
>
> > Question 1: Why does Table 6 evaluate the model's performance only on the sentence fusion task?
>
> A: We supplement the results on WikiSplit and GYAFC datasets. We randomly sample 500 data points for training on LLaMa-7B and G-SPEED respectively. GYAFC-E&M and GYAFC-F&R are the subsets of the GYAFC corpus in the Entertainment & Music and Family & Relationships domains. The evaluation metrics of WikiSplit and GYAFC are SARI and BLEU respectively. The following results still support our original conclusion:
>
> |          | WikiSplit | GYAFC-E&M | GYAFC-F&R |
> | -------- | --------- | --------- | --------- |
> | LLaMa-7B | 50.1      | 44.2      | 46.0      |
> | G-SPEED  | **54.4**  | **46.2**  | **46.6**  |
>
> > Question 2: In Table 3, the performance of w/o ADDI. outperforms G-SPEED on ITR-F and ITR-L. Please explain the reason for this discrepancy.
>
> A: In the Fluency task, the model has a significant improvement on the JFL test set after additional fine-tuning, while the performance on the ITR-F test set is worse. Because the data volume of ITR-F is too small (88), some slight changes will lead to large differences in results. In actual observation, we did not find obvious quality degradation. In ITR-L, we think the quality of the fine-tuning dataset is responsible for this problem. We use the same data to fine-tune LLaMa-7B and the SARI score of LLaMa-7B also drops from 34.1(before training ) to 33.3(after training), which confirms our hypothesis.

---

### Official Review · Reviewer_XAag · 2023-08-05

**Soundness:** 4

**Excitement:**

4: Strong: This paper deepens the understanding of some phenomenon or lowers the barriers to an existing research direction.

**Paper Topic And Main Contributions:**

This paper presents an unsupervised data clustering algorithm for data collection to train editing models, and a sparse BERT-based editing model.

**Questions For The Authors:**

A.	Could you check if there is any contamination between data collected by your clustering algorithm and the test sets of the evaluation tasks?

B.	For an input with a single editing intent (say fluency), could different experts (corresponding to different intents) be activated at different sparse BERT layers? If yes, what does this mean? If we assume single intent input, were any model variations tried to enforce that the same expert is activated at all layers?

C.	In Table 3, the performance of G-SPEED on clarity seems to improve without pre-training or without fine-tuning. Discussion on this could be added.

**Reasons To Accept:**

1.	The unsupervised data clustering algorithm is shown to be effective in training editing models.
2.	The sparse BERT-based editing model when either pre-trained or fine-tuned is shown to have competitive performance with larger LMs used in a prompting-based manner.

**Reasons To Reject:**

1.	Unsure of contamination between data collected in an unsupervised manner and the test set of the evaluation tasks.
2.	Comparisons could be performed with other editing models like Levenshtein Transformer, etc.

**Reproducibility:**

4: Could mostly reproduce the results, but there may be some variation because of sample variance or minor variations in their interpretation of the protocol or method.

**Reviewer Confidence:**

2: Willing to defend my evaluation, but it is fairly likely that I missed some details, didn't understand some central points, or can't be sure about the novelty of the work.

---

> ### Author Rebuttal · Authors · 2023-08-29
>
> We highly appreciate your thoughtful reviews! We hope the following response can address your questions and concerns well.
>
> > Reject Reason 1: Unsure of contamination between data collected in an unsupervised manner and the test set of the evaluation tasks.
> >
> > Question A: Could you check if there is any contamination between the data collected by your clustering algorithm and the test sets of the evaluation tasks?
>
> A: This is a really thoughtful comment. After carefully checking data sources, we think there is no contamination between our pre-training data and the test sets of the evaluation tasks.  Specifically, in the evaluation benchmark, JFLEG, Turk, ASSET, and ITERATER are manually annotated; STSB is collected from forums and news. Only the WNC dataset is collected from the revision history of Wikipedia passages. However, WNC is collected from the revisions made between 2004 and 2019, while our dataset is collected from the revision histories in the March 1st, 2023 dump of English Wikipedia. For further confirmation, we first concatenate the source and target of each data sample and use traversal search to filter the training samples of our pre-training datasets that have a BLUE value of 0.9 with any data sample of WNC. However, no such data sample satisfies this requirement, which further confirms there is no contamination. We will supplement these details in the next version.
>
> > Reject Reason 2: Comparisons could be performed with other editing models like Levenshtein Transformer, etc.
>
> A: Thanks for your suggestions. Because Levenshtein Transformer mainly focuses on machine translation tasks, we select two popular editing models (e.g., LaserTagger [1] and FELIX [2]) as baselines. Below are the SARI scores of different models:
>
> |                 | JFL      | ITR-F    | ITR-L    | ITR-O    | STS      | TRK      | AST      | WNC      |
> | --------------- | -------- | -------- | -------- | -------- | -------- | -------- | -------- | -------- |
> | LaserTagger [1] | 40.0     | 34.1     | 33.0     | 33.1     | 25.6     | 33.1     | 35.7     | 35.1     |
> | FELIX [2]       | 36.1     | 34.2     | 31.5     | 34.8     | 29.1     | 34.5     | 36.3     | 35.2     |
> | G-SPEED         | **54.2** | **42.2** | **34.0** | **41.0** | **36.0** | **38.5** | **40.6** | **60.2** |
>
> We can observe that G-SPEED outperforms LaserTagger and FELIX. We will report these results in the next version.
>
>
>
> >  Question B: For an input with a single editing intent (say fluency), could different experts (corresponding to different intents) be activated at different sparse BERT layers? If yes, what does this mean? If we assume single intent input, were any model variations tried to enforce that the same expert is activated at all layers?
>
> A: Different experts will not be activated at different sparse BERT layers for an input with a single editing intent. The routing method we use in G-SPEED enforces the activation of the same expert at all layers. In the next version, we will revise the Section 5 for more clarity.
>
> > Question C: In Table 3, the performance of G-SPEED on clarity seems to improve without pre-training or without fine-tuning. Discussion on this could be added.
>
> A: Thanks for your suggestions. The result without pre-training is only 0.1 higher than the G-SPEED result in SARI, and GLEU is reduced by 0.3, so the major problem is that the effect of fine-tuning is not good for the pre-training model. We think the quality of the fine-tuning dataset is responsible for this problem. We use the same data to fine-tune LLaMa-7B and the SARI score of LLaMa-7B also drops from 34.1(before training ) to 33.3(after training), which confirms our hypothesis. We will supplement these details in the next version.
>
>
>
> References:
>
> [1] Malmi E, Krause S, Rothe S, et al. Encode, Tag, Realize: High-Precision Text Editing[C]//Proceedings of the 2019 Conference on Empirical Methods in Natural Language Processing and the 9th International Joint Conference on Natural Language Processing (EMNLP-IJCNLP). 2019: 5054-5065.
>
> [2] Mallinson J, Severyn A, Malmi E, et al. FELIX: Flexible Text Editing Through Tagging and Insertion[C]//Findings of the Association for Computational Linguistics: EMNLP 2020. 2020: 1244-1255.

---

### Official Review · Reviewer_qsvy · 2023-08-07

**Typos Grammar Style And Presentation Improvements:** 1. Missing full stop in Line071.

2. …
**Soundness:** 4

**Excitement:**

4: Strong: This paper deepens the understanding of some phenomenon or lowers the barriers to an existing research direction.

**Missing References:**

1. As editing data clustering is one of the major contributions of the paper, it is necessary to discuss relevant editing dataset construction methods in Related Work.

**Paper Topic And Main Contributions:**

This paper targets the task of text editing, where an input document is edited according to a given intent. To tackle the task, this paper proposes G-SPEED, a lightweight general editing model. G-SPEED is made up of a tagging model and a generation model. The tagging model learns to mark the editing operation for each word in the document, while the generation model learns to generate new words or transformed words based on the tagging results in a non-autoregressive manner. To further boost the model performance, this paper employs a mixture-of-expert strategy for both the tagging and generation models. Specifically, each expert is instantiated as an independent FFN in a Transformer block. To pre-train the G-SPEED for general editing, this paper proposes unsupervised editing data clustering, a method to curate datasets of <Text before edited, Text after edited, Edit intent>. To evaluate the performance of G-SPEED, the paper conducts experiments on the EditEval benchmark. Experimental results show that G-SPEED achieves state-of-the-art performance on some editing tasks.

**Questions For The Authors:**

1. The EditEval includes 7 kinds of editing intents. I would like to know the reasons for not choosing the Update editing task.

2. Are there any quality studies about the pre-training dataset?

3. Please discuss the pros and cons of LLM-based generative editing approaches (e.g., ChatGPT) and G-SPEED. It is observed from Table 2 that G-SPEED does not win across the board on EditEval. LLM-based generative editing approaches still outperform 3 of 6 tasks. Also, though G-SPEED is light-weight and powerful, it has limitations in generalizing to unseen editing tasks. To the best of my knowledge, G-SPEED needs to be finetuned with labeled data before being used for new editing tasks, such as the clarity task in EditEval, but ChatGPT like approaches do not have such limitations. Hence, to give comprehensive guidance for the community and practitioners, it is recommended to holistically discuss the pros and cons of G-SPEED and ChatGPT like approaches.

**Reasons To Accept:**

1. The paper is well-written and easy to follow.

2. The proposed clustering method and its resulting editing data is useful for the community to advance the field of text editing. Also, the impressive performance of the light-weight G-SPEED model could benefit more applications and practitioners.

**Reasons To Reject:**

1. Limited technical novelty. The tagging technique, non-autoregressive editing tokens generation, and sparse FFN layer have been explored extensively in the literature. The ablation study and analysis cannot comprehensively tell the reasons why G-SPEED outperforms prior work by a large margin. It is suggested that the authors elaborate the differences with prior work in the Editing Model subsection in Related Work.

**Reproducibility:**

2: Would be hard pressed to reproduce the results. The contribution depends on data that are simply not available outside the author's institution or consortium; not enough details are provided.

**Reviewer Confidence:**

4: Quite sure. I tried to check the important points carefully. It's unlikely, though conceivable, that I missed something that should affect my ratings.

---

> ### Author Rebuttal · Authors · 2023-08-29
>
> Thanks for your reviewing efforts and valuable comments! The following responses are intended to respond to your questions and alleviate your concerns.
>
>
>
> > Reject Reason: Limited Technical Novelty.
>
> A: In this work, to the best of our knowledge, we are the first to propose a lightweight general text editing model (G-SPEED) with only 508M parameters, which can surpass LLMs equipped with 175B parameters. Due to the data scarcity problem of editing tasks and small language models' inherently limited learning capabilities, simply combining existing methods cannot obtain a general editing model based on small language models. Therefore, to meet this challenging goal, apart from testing different variants of existing methods empirically, we propose a novel unsupervised text editing data clustering method and a sparse editing model architecture. Both of these strategies are under-explored in the field of text editing, and the experimental results fully confirm the effectiveness of our proposed methods. Hence, we believe this work makes significant and nontrivial technical contributions to promote the development of text editing.
>
> > Question 1: The EditEval includes 7 kinds of editing intents. I would like to know the reasons for not choosing the Update editing task.
>
> A: Sorry for the missing information. The Update editing task requires references from external sources that are relevant to the particular task.  This task is inconvenient to evaluate because text updates can be multi-directional, especially when we have multiple references without a clear update direction (such as correcting hallucinations) and only one reference answer. One parallel work [1] doesn't choose the Update editing task for the same reason. We will supplement this information in the next version.
>
>
>
> > Question 2: Are there any quality studies about the pre-training dataset?
>
> A: In Table 7, we have shown the center words and comments of each cluster of the pre-training dataset. In the next version, we will further use UMAP to visualize the data clustering results and combine the high-frequency word statistics to analyze the differences and similarities between multiple clusters. We will also conduct more case studies of the pre-training dataset in the next version.
>
>
>
> > Question 3: Please discuss the pros and cons of LLM-based generative editing approaches (e.g., ChatGPT) and G-SPEED.
>
> A: Thanks for your suggestions. In lines 441-464, we have compared the performance of G-SPEED and the LLM-based approaches on different tasks. Following your guidelines, in the next version, we will supplement more discussion of the pros and cons of performance, cost, and generalization. (1). In terms of performance, due to the different requirements of various text editing tasks and the different characteristics of models, G-SPEED and the LLM-based approaches excel in different types of tasks. Specifically, G-SPEED tends to make more precise modifications with caution, while the LLM-based approaches are generated from scratch in an autoregressive manner and hence more likely to encounter over-correction problems; (2) In terms of time and space cost, G-SPEED has an absolute advantage over LLMs. G-SPEED not only calculates fewer parameters but also generates responses in a non-autoregressive way, so it is very suitable as a flexible and lightweight application (3). In terms of generalizing to unseen editing tasks, because the pre-training stage is irrelevant to downstream tasks,  G-SPEED can generalize to unseen editing tasks without the need for additional fine-tuning on downstream tasks. Specifically, in Table 3, we can observe that without additional fine-tuning on the train data of downstream tasks, G-SPEED can still outperform InstructGPT and ChatGPT, which shows the strong generalization ability of G-SPEED to unseen editing tasks.
>
>
>
> Reference:
>
> [1] Zhang Y, Cui L, Cai D, et al. Multi-Task Instruction Tuning of LLaMa for Specific Scenarios: A Preliminary Study on Writing Assistance[J]. arXiv preprint arXiv:2305.13225, 2023.

---

### Meta-Review · Area_Chair_PfRt · 2023-09-16

**Recommendation:** 2

**Metareview:**

The paper "G-SPEED: General Text Editing with Sparse BERT and Mixture-of-Experts" presents a lightweight text editing model called G-SPEED, which is composed of a tagging model and a generation model. The paper's main contributions include the proposal of G-SPEED, the unsupervised data clustering method, and the mixture-of-experts strategy for the tagging and generation models. The strengths of the paper include the well-written and easy-to-follow presentation of the proposed methods, the usefulness of the clustering method and the resulting editing data for advancing the field of text editing, and the impressive performance of the lightweight G-SPEED model. The paper also provides experimental results that demonstrate the effectiveness of G-SPEED, achieving state-of-the-art performance on some editing tasks. However, the reviewers also raise a number of concerns or suggestions to improve this paper. Overall, the technical novelty of the proposed techniques can be discussed further and enhanced. The paper does not provide a comprehensive analysis of the differences with prior work in the Editing Model subsection in Related Work. Secondly, the experimental setting and results analysis can be strengthened. Thirdly, the paper does not provide a comprehensive discussion of the pros and cons of LLM-based generative editing approaches, such as ChatGPT, compared to G-SPEED. Although G-SPEED is lightweight and powerful, it has limitations in generalizing to unseen editing tasks, whereas LLM-based approaches do not have such limitations. Finally, the paper should discuss the potential contamination between the data collected in an unsupervised manner and the test sets of the evaluation tasks, which raises concerns about the validity of the results.

---

### Decision · Program_Chairs · 2023-10-07

**Decision:**

Accept-Findings

**Comment:**

The paper "G-SPEED: General Text Editing with Sparse BERT and Mixture-of-Experts" presents a lightweight text editing model called G-SPEED, which is composed of a tagging model and a generation model. The paper's main contributions include the proposal of G-SPEED, the unsupervised data clustering method, and the mixture-of-experts strategy for the tagging and generation models. The strengths of the paper include the well-written and easy-to-follow presentation of the proposed methods, the usefulness of the clustering method and the resulting editing data for advancing the field of text editing, and the impressive performance of the lightweight G-SPEED model. The paper also provides experimental results that demonstrate the effectiveness of G-SPEED, achieving state-of-the-art performance on some editing tasks. However, the reviewers also raise a number of concerns or suggestions to improve this paper. Overall, the technical novelty of the proposed techniques can be discussed further and enhanced. The paper does not provide a comprehensive analysis of the differences with prior work in the Editing Model subsection in Related Work. Secondly, the experimental setting and results analysis can be strengthened. Thirdly, the paper does not provide a comprehensive discussion of the pros and cons of LLM-based generative editing approaches, such as ChatGPT, compared to G-SPEED. Although G-SPEED is lightweight and powerful, it has limitations in generalizing to unseen editing tasks, whereas LLM-based approaches do not have such limitations. Finally, the paper should discuss the potential contamination between the data collected in an unsupervised manner and the test sets of the evaluation tasks, which raises concerns about the validity of the results.